# ATTENTION-GUIDED DEEP ADVERSARIAL TEMPORAL SUBSPACE CLUSTERING FOR MULTIVARIATE SPATIOTEMPORAL DATA

## ABSTRACT

Deep subspace clustering models provide an efficient solution to the problem of unsupervised subspace clustering of multivariate spatiotemporal data. These clustering solutions are often needed in applications such as snow melt detection, sea ice tracking, crop health monitoring, tracking infectious disease spread, network load prediction, location-based advertising and land-use planning, where multivariate spatiotemporal data exhibit complex temporal dependencies and lie on multiple non-linear manifolds whose internal structure cannot be effectively captured by traditional clustering methods. Existing deep subspace clustering models learn non-linear mappings by projecting data unto a latent space in which data lie in linear subspaces and exploit the self expressiveness property. While this approach has shown impressive performance, they have shortcomings. First, they employ "shallow" autoencoders that completely rely on the self expressiveness of latent features and disregard potential clustering errors. Second, they focus solely on global features while overlooking local features in subspace self-expressiveness learning. Third, they do not capture long-range dependencies or positional information, both of which are crucial for effective spatial and temporal feature extraction and often lead to sub-optimal clustering outcomes. Fourth, their application to 4D multivariate spatiotemporal data remains underexplored. To address these limitations, we propose a novel Attention-Guided Deep Adversarial Subspace Clustering (A-DATSC) model for multivariate spatiotemporal data. A-DATSC incorporates a deep subspace clustering generator and a quality-verifying discriminator that work in tandem. Inspired by the U-Net architecture, the generator preserves the spatial and time-wise structural integrity, reduces the number of trainable parameters and improves generalization through the use of stacked TimeDistributed convLSTM2D layers. It further introduces a graph attention transformer-based self expressive network which captures local spatial relationships, global dependencies and both short and long range correlations crucial for understanding how distant regions and time periods influence each other. When evaluated on three real-world multivariate spatiotemporal datasets, A-DATSC outperforms deep subspace clustering models with significant margins.

## 1 INTRODUCTION

Recent years have seen increased availability in spatiotemporal data from common sources such as government surveys, mobile and wearable devices, launched satellite and weather sensors. These data sources acquire, compress, store, transmit, and process massive amounts of complex high-dimensional multivariate spatiotemporal data. Although this data is high dimensional (Yang et al., 2021), their intrinsic dimension (i.e number of variables needed to describe a data distribution) is often much smaller than the dimension of the ambient space (Pope et al., 2021). For example; in image data processing, the number of pixels in an image can be rather large, yet most image processing models use only a few parameters to describe, for instance the appearance, geometry, and dynamics of a scene. This has motivated the development of techniques like autoencoders and regularization methods (Gonzalez & Balajewicz, 2018; Zhu et al., 2018) for representing high-dimensional data in a lower dimension. Another technique for representing high-dimensional dataset

in a lower dimension is the Principal Component Analysis (PCA) (Kurita, 2019). It assumes that the data is drawn from a single low-dimensional subspace within a high-dimensional space. However, in practice, data points may come from multiple subspaces, and the membership of these points to their respective subspaces is often unknown. This creates a complex sample distribution problem, particularly in multidimensional spatiotemporal data. Therefore, it is necessary to group data points into clusters, where each cluster contains points from the same subspace. This approach assumes that data lies in different subspaces Chen et al. (2020). A category of classical subspace clustering methods have been proposed Chen et al. (2020); Liu et al. (2012); Xu et al. (2021); Ding et al. (2024). A few researchers Yang et al. (2019); Dang et al. (2020); Li et al. (2021); Ji et al. (2017) showed that joint subspace clustering and deep learning have promising performance on benchmark datasets. However, these approaches can hardly be extended to large-scale datasets because they need to learn a self-expressive matrix leading to quadratic time and space complexities. Consequently, some latest works Zhang et al. (2018); Fan (2021); Zhang et al. (2021) dedicate to improving the efficiency of subspace clustering but to the best of our knowledge, there is currently no literature on applying joint subspace clustering and deep learning on multidimensional multivariate spatiotemporal data.

In this paper, we advance research in this area by designing an end-to-end deep temporal subspace clustering model tailored for complex multidimensional multivariate spatiotemporal data. It consists of a deep subspace clustering generator and a quality-verifying discriminator that learns to supervise the generator by evaluating clustering quality in an unsupervised manner. Drawing inspiration from the recent success of the U-net architecture (Ronneberger et al., 2015) in representation learning, our generator incorporates a deep autoencoder composed of stacked convLSTM2D layers and graph attention transformer-based self expressive network and a clustering layer organized in series to capture compact and informative representations of spatial, temporal and salient features of the data. These components capture both local and global patterns, long-range dependencies and positional awareness essential for learning meaningful spatial and temporal patterns and relationships. The clustering layer uses the inherent logic of the Student's t-distribution and iteratively improves clustering result. At the same time, the decoder module adjusts its weights to reduce the disparity between the input and reconstructed data while learning to reconstruct the multidimensional spatiotemporal input data from lower-dimensional latent features. To sum up, this paper makes the following contributions: 1) We propose a novel Attention-guided deep adversarial temporal subspace clustering (A-DATSC) model for 4D multivariate spatiotemporal data. The generator preserves the spatial and time-wise structural integrity, reduces the number of trainable parameters and improves generalization. 2) We design a unified graph attention transformer-based self expressive network which captures local spatial relationships, global dependencies and both short and long range correlations. 3) We design an energy-based, time-varying mini-batch discriminator that leverages temporal subspace modeling to better distinguish between real and fake feature sequences. The remainder of the paper is structured as follows. Section 2 summarizes the background while 3 discusses related works. Section 4 describes the problem in detail while Section 5 presents our proposed solution to the problem. Section 6.3 evaluates results from our proposed model while Section 7 concludes our research.

## 2 Background and Motivation

The exponential growth of multivariate spatiotemporal data across disciplines has created both unprecedented opportunities and formidable analytical challenges. These data are high-dimensional, noisy, heterogeneous, and often exhibit strong nonlinear dependencies across space, time, and variables. Conventional clustering methods, which treat samples as independent and identically distributed (iid), fail to capture these intricate dependencies and often miss the latent low-dimensional subspace structure that governs real-world dynamics. This motivates the pursuit of deep subspace clustering (DSC) methods that can uncover meaningful representations of complex spatiotemporal systems, disentangle overlapping patterns, and group data into coherent clusters that are physically interpretable and temporally consistent. For multivariate spatiotemporal data, these subspaces may represent distinct climate regimes, transportation flow patterns, disease outbreak waves, or other structured phenomena. The development of deep neural architectures particularly those leveraging convolutional, recurrent, and attention-based modules enable learning hierarchical feature representations that preserve spatial locality, model temporal continuity, and capture complex cross-variable correlations. Integrating subspace clustering with representation learning is therefore a powerful

paradigm: it simultaneously discovers a latent feature space and a segmentation of the data into meaningful subspaces, improving robustness to noise and scalability to large datasets.

The motivation for this research is also deeply societal. For instance, in climate science, accurately clustering snowmelt regions, sea-ice zones, or drought-affected areas can improve predictions of sea-level rise, inform resource allocation for adaptation, and guide early warning systems for vulnerable communities. In epidemiology, spatiotemporal clustering can reveal emerging hotspots of disease transmission and support timely interventions. Developing robust, interpretable, and generalizable deep subspace clustering models thus contributes not only to advancing machine learning theory but also to decision support in high-stakes domains where timely insights can save lives, protect infrastructure, and shape policy. Furthermore, research in this area advances the broader field of representation learning by providing a testbed for learning disentangled, causally meaningful embeddings of complex systems. Deep subspace clustering models that are interpretable and explainable have the potential to bridge the gap between data-driven predictions and scientific discovery, enabling domain experts to trust and adopt deep learning in critical workflows. This alignment of methodological innovation, real-world impact, and scientific discovery makes the study of deep subspace clustering of multivariate spatiotemporal data both intellectually compelling and socially urgent.

## 3   RELATED WORK

**Self-expressive learning for deep subspace clustering**. These methods learn self-expression coefficient matrices that capture the relationships between data points. Given a data matrix $X \in \mathbb{R}^{d \times n}$, we express each data point as a linear combination of other data points as: $X = XM$, where $M \in \mathbb{R}^{n \times n}$ is the self-expression coefficient matrix. The optimization problem is: $\min_M \|X - XM\|_F^2 + \lambda \|M\|_1$. Inspired by recent advances in deep learning, Zhang et al., (Zhang et al., 2021) proposed a novel framework for subspace clustering Self-Expressive Network (SENet), which employs two multilayer preceptrons (MLPs) referred to as Query-Net and Key-net to learn a self-expressive representation of the data. While SENet may work well on out of sample data, it struggles to capture long-range dependencies and positional awareness, a vital component for subspace clustering of multivariate spatiotemporal data. Recently, Baek et al., (Baek et al., 2021) proposed Deep self-representative subspace clustering network for unsupervised subspace clustering to improve representativeness and clustering ability. Although they attempt to improve clustering ability, they completely rely on self expression as supervision and do not preserve local features or geometric relationships between data point. Recently Zhao et al., (Zhao et al., 2023) proposed a double self-expressive subspace clustering algorithm which improves performance by preserving the structural information in the self-expressive coefficient matrix.

**Adversarial Networks for subspace clustering**. Recently, there is growing interests in combining the strengths of GANs with subspace clustering methods to enhance clustering performance in complex high-dimensional datasets. Zhou et al. proposed a Deep Adversarial Subspace Clustering (DASC) (Zhou et al., 2018) which introduces adversarial learning and supervises the generator's learning to produce more favorable representations for better subspace clustering. While they addressed the clustering error with little reliance on self-expression for supervision, they overlooked local features, useful long-range dependencies and positional information in feature representation. Mukherjee et al., (Mukherjee et al., 2019) proposed clusterGAN and demonstrated that while one can potentially exploit the latent-space back-projection in GANs to cluster, the cluster structure is not retained in the GAN latent space. Recently, Yu et al., (Yu et al., 2020) proposed two GAN-based enhanced deep subspace clustering approaches: deep subspace clustering via dual adversarial generative networks (DSC-DAG) and self-supervised deep subspace clustering with adversarial generative networks ($S^2$ DSC-AG) and use adversarial training to simultaneously learn the distributions of both the inputs and latent representations.

**Deep Learning based Clustering.** The limitations of traditional clustering methods have motivated the development of deep learning-based approaches, which are better equipped to model nonlinear, high-dimensional data. Deep Embedded Clustering (DEC) Xie et al. (2016) introduced the paradigm of jointly learning representations and cluster assignments by minimizing a Kullback–Leibler (KL) divergence loss between predicted and target distributions. Extensions of DEC and related autoencoder-based methods have been applied to time series data, though many approaches either focus solely on temporal patterns or image-level spatial similarities, neglecting

the joint spatiotemporal structure. To address this gap, spatiotemporal autoencoders Faruque et al. (2023) have emerged, combining convolutional neural networks (CNNs) with recurrent architectures such as Long Short-Term Memory (LSTM) networks, proving highly effective for spatiotemporal data, as it integrates convolutional operations into recurrent units, allowing the model to simultaneously capture localized spatial patterns and their evolution over time. This approach has been successfully applied to applicatins such as precipitation nowcasting Shi et al. (2017), sea ice prediction in the Arctic Wang et al. (2019), and regional climate variability detection Liu et al. (2020), demonstrating its capability to extract meaningful representations from complex spatiotemporal climate datasets. These applications highlight the model's ability to capture both fine-scale spatial correlations and their evolution across temporal sequences. Recent advances in graph neural networks (GNNs) have opened new opportunities for modeling dynamic spatial and temporal relationships. Temporal Graph Attention Networks (TGAT) Xu et al. (2020) extend graph convolution by incorporating temporal attention, enabling models to learn evolving spatiotemporal interactions while adaptively weighting neighbors based on their temporal relevance. In geoscience, GNNs and TGAT variants have been explored for applications such as air quality forecasting Jiang et al. (2023), urban climate modeling Li et al. (2023), and climate teleconnection discovery Peng et al. (2021).

## 4 PROBLEM DEFINITION

Let $U = \bigcup_{i=1}^{M} S_i$ be the nonlinear set consisting of a union of $M$ subspaces $\{S_i \subset H\}_{i=1}^{M}$, where $S_i$ are subspaces of a Hilbert or a Banach space $H$. Let $W = \{w_j \in H\}_{j=1}^{N}$ be a set of multivariate spatiotemporal data points drawn from $U$. Each data point $w_j$ is represented as a high-dimensional tensor $w_j \in \mathbb{R}^{T \times \text{lon} \times \text{lat} \times \text{var}}$, where $T$ denotes the temporal dimension, and *lon, lat* denote the spatial dimensions across multiple variables *var*.

**Goal:** Our objective is to segment this temporal sequence into $K$ coherent clusters $\{\mathcal{C}_1, \mathcal{C}_2, \ldots, \mathcal{C}_K\}$ such that time steps within the same cluster share similar *latent subspace representations* that capture both their spatial structure and multivariate interactions. For a predefined number of subspaces $M$ with intrinsic dimensions $\{d_i\}_{i=1}^{M}$, the problem can be formalized as the following minimization: $e(W, S) := \sum_{f \in W} \min_{1 \le j \le M} d_{\mathcal{H}}(f, S_j)$, where $S = \{S_1, \ldots, S_M\}$ is a candidate set of subspaces, $d_{\mathcal{H}}(\cdot, \cdot)$ is the distance induced by the norm on $\mathcal{H}$, and $e(W, S)$ measures the total reconstruction error. The task is to find $S^* = \{S_1^*, \ldots, S_M^*\} = \arg\min_{S \in \mathcal{S}} e(W, S)$.

*Learning Orthonormal Bases.* For each subspace $S_i$, we seek an orthonormal basis $\{u_{i1}, \ldots, u_{id_i}\} \subset S_i$, such that $S_i = \text{span}\{u_{i1}, \ldots, u_{id_i}\}$, $\langle u_{ij}, u_{ik} \rangle_{\mathcal{H}} = \delta_{jk}$, where $d_i = \dim(S_i) \ll \dim(\mathcal{H})$ is the intrinsic dimension, $\langle \cdot, \cdot \rangle_{\mathcal{H}}$ is the inner product in $\mathcal{H}$, and $\delta_{jk}$ is the Kronecker delta. In deep subspace clustering, these basis vectors are learned implicitly via a deep encoder $f_\theta$ producing latent representations: $z_j = f_\theta(w_j) \in \mathbb{R}^d$. Points from the same subspace are expected to lie near a linear subspace in $\mathbb{R}^d$, allowing PCA or SVD to recover an orthonormal basis for each subspace. *Clustering Assignment:* Define a clustering function $\varphi : W \to \{1, \ldots, M\}$, such that each $w_j \in W$ is assigned to a unique subspace $S_{\varphi(w_j)}$. The resulting clusters are $C_i = \{w_j \in W \mid \varphi(w_j) = i\}$, $i = 1, \ldots, M$. *Clustering Constraints.* The clusters must satisfy: 1) *Partition:* $\bigcup_{i=1}^{M} C_i = W$, 2). *Disjointness:* $C_i \cap C_j = \emptyset$, $\forall i \ne j$, and 3) *Subspace Membership:* $C_i \subset S_i \subset \mathcal{H}$ for each $i$.

## 5 METHODOLOGY

In this section, we present the architecture and training strategy of the proposed Attention-guided deep adversarial temporal subspace clustering (A-DATSC) model. As shown in Figure 1, A-DATSC couples a spatiotemporal encoder-decoder with a bidirectional temporal graph attention bottleneck, a DEC-style per-timestep clustering head with temperature and balancing regularizers, a self-expressive temporal layer in the generator $G$, and an energy-based subspace discriminator $D$.

### 5.1 GENERATOR ($G$)

$G$ is designed to jointly learn hierarchical spatiotemporal representations, a causally informed temporal affinity structure, and a clustering assignment that is robust, balanced, and interpretable. $G$ is composed of four key components: (i) a ConvLSTM-based spatiotemporal encoder, (ii) a bidirec-

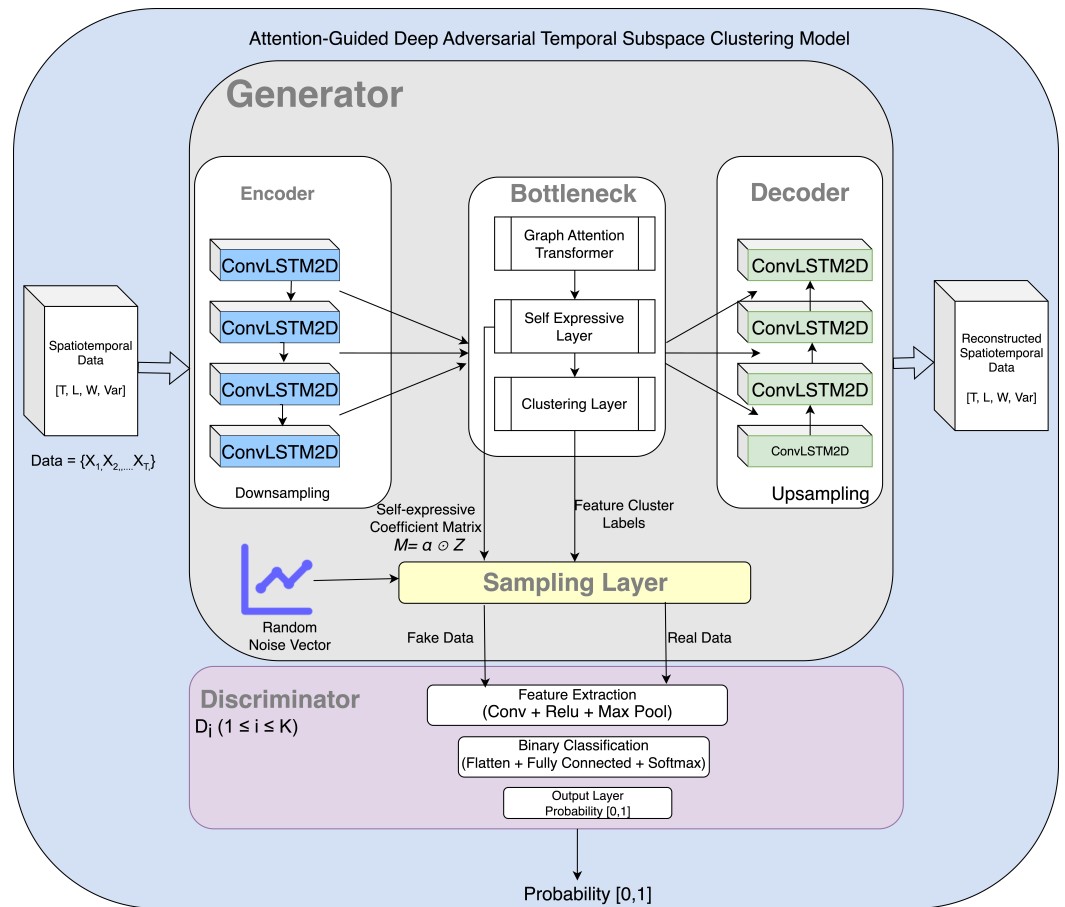

Figure 1: Architecture of Attention-guided deep adversarial temporal subspace clustering (A-DATSC) model. A-DATSC is composed of a deep subspace and clustering generator and a quality-verifying discriminator. The generator is composed of an autoencoder and a sampling layer. The autoencoder receives 4D multivariate spatiotemporal data as input and outputs cluster labels, self-expressive coefficient matrix and reconstructed data. The sampling layer receives as input the cluster labels, coefficient matrix and random noise vectors and outputs real clustered data features and fake data features. Both feature vectors are sent to the discriminator to determine real or fake data features while generating new data.

tional temporal graph attention transformer(BiTGAT) layer, (iii) a self-expressive temporal subspace layer, and (iv) a decoder with skip connections to ensure faithful reconstruction. Figure 1 provides a high-level schematic of the end-to-end pipeline.

**Notation and Input Representation:** Let the input be a multivariate spatiotemporal tensor $\mathbf{X} \in \mathbb{R}^{B \times T \times H \times W \times C}$, with batch size $B$, time steps $T$, spatial grid $H \times W$, and $C$ variables. We write $\mathbf{X}_b$ for the $b$-th sample, and $\mathbf{X}_{t,:,:,:} \in \mathbb{R}^{H \times W \times C}$ for the $t$-th frame of a sequence. Our goal is to produce per-timestep soft cluster assignments $\mathbf{q}_t \in \Delta^{K-1}$ over $K$ clusters and to learn subspace-aware embeddings $\mathbf{z}_t \in \mathbb{R}^D$ that are discriminative, temporally coherent, and subspace-preserving.

### 5.1.1 SPATIOTEMPORAL ENCODER (CONVLSTM WITH RESIDUAL TEMPORAL BLOCKS):

The encoding phase follows a U-Net downsampling structure composed of `TimeDistributed ConvLSTM2D` layers with spatial max pooling. This encoder is responsible for learning low-dimensional latent representations that capture both spatial and temporal correlations. By applying convolutions across both space and time, the encoder compresses the input data into a bottleneck representation $Z \in \mathbb{R}^{T \times d}$, where $T$ denotes the temporal dimension and $d$ the latent feature di-

mensionality. For brevity, let $\{X_1 \ldots, X_n\}$ denote the input samples and let $\{z_1 \ldots, z_n\}$ denote their corresponding latent representations learned by the encoder in $G$. Namely, $z_i \in \mathbb{R}^d$ is the d-dimensional representation of the $i-$th 4D sample $X_i \in \mathbb{R}^{T \times H \times W \times C}$ and $k$ denotes the number of clusters or subspaces. The mapping from an input data space to the latent feature space is a non-linear function $f_{\text{enc}} := X \to Z$, were $Z \in \mathbb{R}^m$ is an $m$-dimensional high-level representation of all the variables at each timestep. $Z_t = f_{\text{enc}}(X_t)$, $t = \{1, \ldots, T\}$. From Figure 1, to extract spatial, temporal and salient features at different scales and reduced dimensionality, the encoder applies a ConvLSTM2D stem followed by residual temporal blocks and temporal-preserving spatial pooling:

$$\mathbf{H}^{(1)} = \text{ConvLSTM2D}_{64}(\mathbf{X}); \quad \mathbf{H}^{(l)} = \text{ResTempBlock}_{F_l}\big(\text{MaxPool3D}_{(1,2,2)}(\mathbf{H}^{(l-1)})\big), \quad (1)$$

for $l = 2, 3, 4$ with filters $F_l \in \{128, 256, 512\}$. Each residual temporal block stacks two Con-vLSTM2D layers with LayerNorm and a $1 \times 1 \times 1$ projection if the channel dimension changes. This preserves long-range temporal dependencies while hierarchically compressing spatial resolution ($H/8 \times W/8$ at the top level) and increasing channel capacity (e.g., 512).

**Patchification for Graph Efficiency.** To form a tractable spatiotemporal node set for attention, we tile the top-level tensor into non-overlapping patches of size $(h_p, w_p)$, yielding a reduced grid $(H', W')$ with $N = H'W'$ nodes per frame. We flatten $(H', W')$ to a node axis so that the sequence becomes $\mathbf{X}^{(4)} \to \tilde{\mathbf{X}} \in \mathbb{R}^{B \times T \times N \times F}$, with $F = 512$. Patchification reduces graph size and stabilizes attention training while preserving local spatial structure.

### 5.1.2 Bidirectional Temporal Graph Attention (Bi-TGAT) Bottleneck:

We process the node sequences with a bidirectional temporal GAT layer that aggregates information both forward and backward in time. Let $\mathbf{H}_t \in \mathbb{R}^{N \times F}$ be node features at time $t$. A temporal adjacency (implicit, learned) is built by attention over $\mathbf{H}_t$ and $\mathbf{H}_{t\pm1}$; for each direction $d \in \{\to, \leftarrow\}$ we compute:

$$\alpha_{t,i \to j}^{(d)} = \text{softmax}_j \left( \phi\big(\mathbf{W}_q^{(d)} \mathbf{h}_{t,i}, \mathbf{W}_k^{(d)} \mathbf{h}_{t+\delta_d, j}\big) \right), \quad \delta_\to = +1, \ \delta_\leftarrow = -1, \quad (2)$$

$$\mathbf{m}_{t,i}^{(d)} = \sum_j \alpha_{t,i \to j}^{(d)} \mathbf{W}_v^{(d)} \mathbf{h}_{t+\delta_d, j}, \qquad \mathbf{h}_{t,i}' = \text{LN}\Big(\mathbf{W}_o[\mathbf{m}_{t,i}^{(\to)} \| \mathbf{m}_{t,i}^{(\leftarrow)}]\Big). \quad (3)$$

Multi-head attention stabilizes learning; the outputs are pooled across $N$ nodes to obtain a per-timestep embedding $\mathbf{z}_t \in \mathbb{R}^D$ (via head concat + projection), and globally pooled across $t$ to get a sequence summary $\bar{\mathbf{z}} \in \mathbb{R}^D$. A linear projection and LayerNorm yield the final temporal sequence embeddings $\{\mathbf{z}_t\}_{t=1}^T$. By integrating Bi-TGAT, we fuse local spatial context with directed temporal cues ($t \pm 1$), mitigating exposure bias and enhancing discriminability of transient regimes. Attention weights act as data-adaptive temporal edges, improving subspace separation by emphasizing causally or dynamically influential frames.

### 5.1.3 Per-Timestep Clustering Head with Temperature and Balance

We adopt a DEC-style Student-$t$ assignment per time step (Xie et al., 2016):

$$q_{t,k} \propto \left(1 + \frac{\|\mathbf{z}_t - \boldsymbol{\mu}_k\|_2^2}{\alpha}\right)^{-\frac{\alpha+1}{2}}, \quad \sum_{k=1}^K q_{t,k} = 1, \quad (4)$$

with trainable centers $\{\boldsymbol{\mu}_k\}_{k=1}^K$ and dof $\alpha$. We introduce a temperature $\tau$ to control sharpness: $\tilde{q}_{t,k} \propto q_{t,k}^{1/\tau}$, annealing $\tau \downarrow$ to harden assignments over training. To avoid mode collapse, we combine two balancing terms: (i) a batch-wise KL divergence to a near-uniform marginal to enforce cluster utilization, and (ii) a mutual-information style redundancy penalty across clusters. The head optimizes the classic DEC target distribution $\mathbf{p}$ computed from $\mathbf{q}$ and adds $\text{KL}(\mathbf{p}\|\mathbf{q})$ to the loss. DEC sharpening aligns the extracted features $\mathbf{z}_t$ with centers, while temperature scheduling prevents early overcommitment. Balanced assignments keep clusters populated and reduce mode collapse.

### 5.1.4 Self-Expressive Temporal Layer (SE-T1)

We incorporate a per-timestep self-expressive module guided by the current soft assignments $\tilde{\mathbf{q}}_t$ to emphasize subspace structure. Let $\mathbf{Z} \in \mathbb{R}^{T \times D}$ stack embeddings. We learn a coefficient matrix

$\mathbf{C} \in \mathbb{R}^{T \times T}$ (diagonal masked if exclude-self) via a shrinkage operator so that

$$\mathbf{Z}_{\text{SE}} = \mathbf{CZ}, \quad \mathcal{L}_{\text{SE}} = \|\mathbf{Z} - \mathbf{CZ}\|_F^2 + \lambda_{\text{SE}}\|\mathbf{C}\|_1, \tag{5}$$

optionally weighting entries by temporal proximity and assignment similarity, e.g., $w_{t,s} = \tilde{\mathbf{q}}_t^\top \tilde{\mathbf{q}}_s \cdot \exp(-|t - s|/\sigma_t)$. A soft-threshold (shrink) encourages sparsity; the output $\mathbf{Z}_{\text{SE}}$ is time-averaged and concatenated with the Bi-TGAT pooled vector to form the bottleneck. Self-expression promotes subspace-preserving affinities: each $\mathbf{z}_t$ is reconstructed by a small set of neighbor frames from the same latent subspace, improving block-diagonality of the affinity and boosting spectral separability.

### 5.1.5 DECODER AND RECONSTRUCTION LOSS

The decoder mirrors the encoder with ConvLSTM2D up-paths and skip connections from the encoder stages (temporal up-sampling aligns skip timings). The reconstruction loss $\mathcal{L}_{\text{rec}} = \|\hat{\mathbf{X}} - \mathbf{X}\|_2^2$ anchors the representation to physically plausible spatiotemporal fields, improving stability of the latent space.

Another important function of $G$ is to generate **real** and **fake** samples conditioned on the cluster $C_i$ where $i = 1, \ldots, K$, implemented by the sampling layer. Our discriminator is designed to learn a linear subspace $S_i$ to fit the intrinsic ground-truth subspace $S_i^*$ of cluster $C_i$. Then according to the projection residuals of data points on their corresponding subspaces learned by the discriminator, the discriminator can identify whether the input data are real or fake.

**The Sampling Layer**: The generator additionally produces *real* and *fake* samples per cluster $C_i$ using a reparameterization trick (Kingma & Welling, 2013): $\bar{\mathbf{z}}_t = \sum_{j=1}^{m_i} \alpha_{tj}\mathbf{z}_{ij}, \quad t = 1, \ldots, m_i^*$, where $\alpha_{tj} \sim \mathcal{U}(0, 1)$ are fixed during training, allowing gradients to flow through $\mathbf{z}_{ij}$.

### 5.2 ENERGY-BASED SUBSPACE DISCRIMINATOR

To explicitly enforce linear subspace geometry in latent space, we employ an energy-based discriminator with one subspace basis per cluster. For cluster $k$, learn $\mathbf{U}_k \in \mathbb{R}^{D \times r}$ (column-orthonormal ideally). The projection residual energy of $\mathbf{z}$ onto subspace $k$ is $\mathcal{E}(\mathbf{z}; \mathbf{U}_k) = \left\| \mathbf{z} - \mathbf{U}_k\mathbf{U}_k^\top\mathbf{z} \right\|_2^2$. With current assignments, we sample *real* latent points per cluster (highest responsibilities) and synthesize *fake* latents as convex combinations of in-cluster points using the soft weights. The discriminator minimizes a hinge objective encouraging real energies to be below fake energies by margin $m$:

$$\mathcal{L}_{\text{D}} = \mathbb{E}_{\text{real}}\big[\max\big(0, \mathcal{E}(\mathbf{z}_{\text{real}}; \mathbf{U}) - \overline{\mathcal{E}}(\mathbf{z}_{\text{fake}}; \mathbf{U}) + m\big)\big] + \beta_\perp \sum_{k=1}^K \big\|\mathbf{U}_k^\top\mathbf{U}_k - \mathbf{I}\big\|_F^2 + \beta_\times \sum_{\substack{i,j=1 \\ i \neq j}}^K \big\|\mathbf{U}_i^\top\mathbf{U}_j\big\|_F^2$$

The generator is adversarially trained to *reduce* fake residuals: $\mathcal{L}_{\text{adv}} = \mathbb{E}_{\text{fake}} \mathcal{E}(\mathbf{z}_{\text{fake}}; \mathbf{U})$. The discriminator shapes latent geometry to be union-of-subspaces: low residual within the correct cluster subspace and high residual otherwise. Orthogonality/separation regularizers improve cluster identifiability and reduce overlap between subspaces, increasing temporal cluster purity.

### 5.3 OVERALL OBJECTIVE

The training objective sums reconstruction, clustering, balancing, self-expression, and adversarial terms:

$$\mathcal{L} = \underbrace{\mathcal{L}_{\text{rec}}}_{\text{AE recon}} + \underbrace{\text{KL}(\mathbf{p} \| \mathbf{q})}_{\text{DEC}} + \lambda_{\text{bal}}\Big( \underbrace{\text{KL}(\bar{\mathbf{q}} \| \mathbf{u})}_{\text{marginal balance}} + \underbrace{\mathcal{L}_{\text{MI}}}_{\text{redundancy}} \Big) + \lambda_{\text{SE}}\,\mathcal{L}_{\text{SE}} + \lambda_{\text{adv}}\,\mathcal{L}_{\text{adv}}. \tag{6}$$

Here $\bar{\mathbf{q}}$ is the batch-average assignment, $\mathbf{u}$ is the uniform distribution, and $\mathcal{L}_{\text{MI}}$ penalizes degenerate mutual information across clusters (implementation via contrastive or covariance de-correlation). Temperature $\tau$ is annealed and the balancing ramp is increased during training.

### 5.4 OPTIMIZATION AND TRAINING SCHEDULE

1. **Generator step.** Forward once to obtain $\hat{\mathbf{X}}$, $\{\mathbf{z}_t\}_{t=1}^T$, $\{\mathbf{q}_t\}_{t=1}^T$. Compute reconstruction and clustering objectives $\mathcal{L}_{\text{rec}}$, $\text{KL}(\mathbf{p}\|\mathbf{q})$, balancing loss, and self-expression loss. Synthesize fake latents and compute the adversarial term $\mathcal{L}_{\text{adv}}$. Update encoder/decoder, Bi-TGAT, clustering centers, and SE parameters.

2. **Discriminator step.** Sample real latents per cluster (top-$m$ by responsibility) and synthesize fakes. Update $\{\mathbf{U}_k\}_{k=1}^{K}$ by minimizing $\mathcal{L}_\mathrm{D}$.

3. **Schedules.** Initialize with a larger temperature $\tau$ (softer assignments) and anneal $\tau \downarrow$ over training; ramp balancing coefficients; optionally enable SE after a warm-up phase to avoid early sparse overfitting.

**Inference and Final Clustering** At test time, we compute $\{\mathbf{z}_t\}$ and $\{\mathbf{q}_t\}$. Final hard labels are $\hat{y}_t = \arg\max_k q_{t,k}$. Optionally, we build an affinity $\mathbf{A} = |\mathbf{C}| + |\mathbf{C}^\top|$ from SE coefficients and apply spectral clustering to refine temporal segments, leveraging the induced block-diagonal structure.

## 6 EXPERIMENT

All models are executed on AWS cloud environment using 20GB of S3 storage with 30 GB of ml.g4dn.xlarge GPU. The hardware used is a macOS Sonoma version 14.4.1, 16 GB, M1 pro chip. We applied the same python library across all models for homogeneity. We aim to implement our proposed model using python's machine and deep learnings libraries including Keras 2.11 and TensorFlow 2. All the baseline models and proposed models would be tested on Google Colab notebook with 12 GB GPU - A100, High RAM memory support. The hardware we would be using is a macOS ventura version 13.3, 16 GB, M1 pro chip.

### 6.1 DATASET AND DATA PREPROCESSING

To ensure generalizability, we experimented with three multivariate spatiotemporal datasets: C3S Arctic Regional Reanalysis (CARRA) dataset (Copernicus Climate Change Service (C3S)), European Centre for Medium-Range Weather Forecasts (ECMRWF) ERA-5 global reanalysis product (ECMWF, Copernicus Climate Change Service, 2021), and daily atmospheric observations (NCEP/NCAR). These datasets are provided alongside our implementation code and publicly available. All datasets follow the same preprocessing steps. All three data sets consists of daily observations over the course of one year and presented in four dimensions: longitude, latitude, time, and variables. Our proposed model accepts 4D data but to obtain a dimension suitable for our benchmark models, we transform the data from 4D to 2D tabular data $[time, (lon, lat, var)]$ Existing null values are replaced by the overall mean of the dataset. We apply standard Min-Max Normalization which rescales all features to fall within the range of $[0, 1]$.

### 6.2 BASELINE METHODS

We compare our proposed model against state-of-the-art deep clustering models. These include (DEC) (Xie et al., 2016), (DSC) (Faruque et al., 2023), ClusterGAN Mukherjee et al. (2019), Info-GAN (Chen et al., 2016), (DTC) (Sai Madiraju et al., 2018) and DASC Zhou et al. (2018), Deep Subspace Clustering(DSC-Net-$L_2$) (Ji et al., 2017), and (DSC-DAG)(Yu et al., 2020) respectively. Based on the elbow method, we used $k = 7$ for ERA5 and NCAR datasets and $k = 7$ for CARRA datasets in our experiments.

### 6.3 EVALUATION METRICS

In the absence of ground truth, we evaluate the performance of our proposed model on six internal cluster validation measures: Silhouette Score (Shahapure & Nicholas, 2020), Davies-Bouldin score (DB) (Ros et al., 2023), Calinski-Harabas score (CH) (Wang & Xu, 2019), Average inter-cluster distance (I-CD) (Everitt et al., 2011), Average Variance (Variance) (Montgomery & Runger, 2010) and Average root mean squared error (RMSE) (Willmott & Matsuura, 2005). These measures seek to balance the *compactness* and the *separation* of formed clusters through minimizing intra-cluster distance and maximizing the inter-cluster distance respectively.

### 6.4 EXPERIMENT RESULTS

Table 1 presents our performance results based on selected internal cluster validation measures when applied to all three datasets. On ERA5 data, A-DATSC outperformed all baseline models as reported

by Silhouette, DB, RMSE and I-CD. This implies A-DATSC was able to capture the underlying complex patterns in all three datasets with significant improvements on performance.

Table 1: Performance evaluation of our proposed model: Each selected model is evaluated on all six metrics. Best results is underlined

| | | Baseline Models | | | | | Proposed |
|---|---|---|---|---|---|---|---|
| | Performance | ClusterGAN | DTC | DSC | DEC | DASC | A-DATSC |
| ERA5 | Silhouette ↑ | -0.0989 | 0.2284 | 0.2903 | 0.2050 | 0.1355 | **0.3268** |
| | DB ↓ | 17.1624 | 1.8517 | 1.6741 | 1.7515 | 2.0325 | **1.5009** |
| | CH ↑ | 1.2348 | 72.3222 | **102.2887** | 99.3082 | 72.9257 | 98.8211 |
| 7 optimal | RMSE ↓ | 22.2032 | 15.0820 | 13.6154 | 13.7425 | 15.0477 | **13.5158** |
| clusters | Variance ↓ | 0.1064 | *0.0450* | 0.1033 | *0.0450* | 0.1039 | 0.1038 |
| | I-CD ↑ | 4.0315 | 6.4448 | 6.8481 | 6.8093 | 5.5229 | **7.4839** |
| | | | | | | | |
| | Silhouette ↑ | -0.0753 | 0.0220 | 0.2437 | 0.2027 | -0.1059 | **0.2767** |
| CARRA | DB ↓ | 7.7668 | 2.2332 | 1.6844 | 1.6781 | 11.7039 | **1.5089** |
| | CH ↑ | 4.9468 | 55.5673 | **78.7826** | 68.0469 | 17.4001 | 69.7729 |
| 5 optimal | RMSE ↓ | 7.9781 | 7.0421 | 5.8789 | **5.5029** | 11.3033 | 5.5424 |
| clusters | Variance ↓ | 0.0021 | 0.0016 | **0.0011** | 0.0160 | 0.0011 | 0.0105 |
| | I-CD ↑ | 2.1073 | 2.3191 | 2.5712 | 2.8264 | **3.1404** | 3.0912 |
| | | | | | | | |
| NCAR | Silhouette ↑ | -0.2659 | 0.6230 | 0.61563 | 0.6454 | 0.1132 | **0.6541** |
| Reanalysis 1 | DB ↓ | 9.3799 | **0.7570** | 0.7804 | 0.7639 | 2.0879 | 0.7612 |
| | CH ↑ | 3.3104 | 864.1750 | 862.3665 | 839.4187 | 192.6249 | **868.7555** |
| 7 optimal | RMSE ↓ | 12.1719 | 3.1380 | 3.1410 | 3.1809 | 6.0048 | 3.1180 |
| clusters | Variance ↓ | 0.1770 | 0.1770 | 0.1770 | 0.1770 | 0.1770 | 0.1770 |
| | I-CD ↑ | 0.7357 | 0.8603 | 0.8745 | **0.9465** | 4.0097 | 0.9098 |

## 6.5 ABLATION STUDY

Table 2: Ablation Study

| Performance - based | | | | | |
|---|---|---|---|---|---|
| | Silhouette ↑ | DB ↓ | CH ↑ | RMSE ↓ | Variance ↓ | ICD ↑ |
| A-DATSC$_{sel}$ | 0.2124 | 2.1486 | 84.6793 | 14.4220 | 0.1038 | 6.1603 |
| A-DATSC$_{cnn-lstm}$ | 0.1965 | 1.9576 | 91.9739 | 14.0709 | **0.1032** | 5.6349 |
| A-DATSC$_{gat}$ | 0.2827 | 1.6941 | **99.9202** | 13.716 | 0.1035 | 6.2759 |
| A-DATSC | **0.3268** | **1.5009** | 98.8211 | **13.5158** | 0.1038 | **7.4839** |

We show the importance of various subcomponents of A-DATSC when evaluated on ERA5 Data. A-DATSC$_{sel}$ represents a variant of A-DATSC with only the self-expressive network at the bottleneck. A-DATSC$_{cnn-lstm}$ is a variant without the time distributed integrated ConvLSTM2d. This uses the traditional cnn blocks followed by an LSTM unit. A-DATSC$_{gat}$ is a variant of the A-DATSC with the Bi-TGAT at the dense layer.

## 7 CONCLUSION

In this study, we propose a novel unsupervised Attention-guided deep adversarial temporal subspace clustering (A-DATSC) model capable of clustering 4D high-dimensional multivariate spatiotemporal data. The model adopts adversarial learning with focus on the spatial, temporal and salient features to effectively supervise sample representation learning and subspace clustering. The generator learns a fine-level latent representation of the data, effectively clusters the latent subspace through a graph based self expressive network and clustering layer and finally generates new random samples with similar cluster patterns. The discriminator evaluates the clustering performance and feeds back the information to the generator to help it produce better sample latent representations and subspace clustering. For future work, we plan to improve the model performance by introducing Feature Matching and One-sided label smoothing.

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
