# OpenReview forum: "ATTENTION-GUIDED DEEP ADVERSARIAL TEMPORAL SUBSPACE CLUSTERING FOR MULTIVARIATE SPATIOTEMPORAL DATA"
_ICLR.cc/2026/Conference — ICLR 2026 Conference Withdrawn Submission_

### Official Review · Reviewer_mBhf · 2025-10-18

**Soundness:** 2
**Presentation:** 2
**Contribution:** 1
**Rating:** 2
**Confidence:** 3

**Summary:**

The paper proposes A-DATSC, an attention-guided deep adversarial temporal subspace clustering model for multivariate spatiotemporal data. The generator is a ConvLSTM-based U-Net with skip connections, followed by (i) a bidirectional temporal graph attention (Bi-TGAT) bottleneck, (ii) a DEC-style Student-t clustering head with temperature annealing and balancing regularizers, and (iii) a self-expressive temporal layer; a decoder reconstructs inputs. A separate energy-based subspace discriminator learns a basis per cluster and enforces a union-of-subspaces geometry adversarially. Claimed contributions are: (1) an attention-guided adversarial temporal subspace model for 4D (lon, lat, time, variables) data, (2) a unified graph attention transformer-based self-expressive network capturing local/global and short/long-range correlations, and (3) an energy-based time-varying mini-batch discriminator for temporal subspaces.

**Strengths:**

- The particular combination like ConvLSTM U-Net + Bi-TGAT bottleneck + self-expression + energy-based subspace discriminator with DEC-style clustering is a coherent synthesis targeted at 4D spatiotemporal data. While each ingredient exists, the authors’ claimed novelty lies in how they are glued together.
- The paper provides a clear architectural overview (Figure 1) and a step-by-step description of each module (encoder, Bi-TGAT, clustering head, self-expression, discriminator).

**Weaknesses:**

1. Ambiguous novelty beyond a lego-like composition of prior techniques.
- Adversarial subspace discriminator: Energy-based adversarial training for deep subspace clustering has precedence. The paper’s discriminator resembles learned subspace bases with projection-residual energies and a margin objective, but it is not clear what new theoretical or algorithmic property distinguishes it beyond the union-of-subspaces hinge/orthogonality regularization given.
  - Zhou, Pan, Yunqing Hou, and Jiashi Feng. “Deep adversarial subspace clustering.” Proceedings of the IEEE Conference on Computer Vision and Pattern Recognition. 2018.
- Clustering head: The Student-t assignment with target-distribution sharpening is directly in the DEC lineage. The added temperature/balance is incremental and could be framed as a known stabilization strategy.
  - Xie, Junyuan, Ross Girshick, and Ali Farhadi. “Unsupervised deep embedding for clustering analysis.” International Conference on Machine Learning. PMLR, 2016.
- Temporal graph attention: The Bi-TGAT layer is presented as a bottleneck. However, the work does not articulate why this specific temporal attention instantiation is crucial for subspace preservation relative to other temporal transformers.
  - Nji, Francis Ndikum, Vandana Janaja, and Jianwu Wang. “B-TGAT: A Bi-directional Temporal Graph Attention Transformer for Clustering Multivariate Spatiotemporal Data.” arXiv preprint arXiv:2509.13202 (2025).

2. Ablation is narrow: Only three toggles are explored (self-expression only; CNN-LSTM; GAT) and only on ERA5. Key elements like adversarial term, temperature/balancing, spectral refinement are not ablated. No multi-seed variance or cross-dataset ablation is provided.

**Questions:**

1. What is the precise novelty of the discriminator? Can you clarify how your energy-based subspace discriminator differs from prior adversarial subspace clustering (beyond using a hinge margin and orthogonality penalties)?
2. Since baselines are fed 2D tabularized data while A-DATSC ingests full 4D, can you provide (a) 4D-aware baseline variants (e.g., ConvLSTM/Transformer baselines without adversarial/self-expression) and/or (b) a 2D-A-DATSC variant to equalize inputs? This will clarify where gains come from.
3. Ablations to isolate contributions. Please add ablations for: removing adversarial loss, removing temperature/balancing, disabling self-expression but keeping spectral refinement, and replacing Bi-TGAT with a simple temporal transformer. Include ablations across all three datasets with multi-seed runs.

---

### Official Review · Reviewer_oAor · 2025-10-29

**Soundness:** 2
**Presentation:** 3
**Contribution:** 2
**Rating:** 4
**Confidence:** 4

**Summary:**

The paper attempts to design a deep temporal subspace clustering approach for multi-dimensional multivariate spatiotemporal data. To be specific, the paper presents a so-called attention guided deep adversarial temporal subspace clustering (A-DATSC) approach for spatiotemproal data segmentation. More specifically, A-DATSC builds on a deep autoencoder framework, which consists of a stacked convLSTM2D layers as a spatiotemporal autoencoder, a graph attention transformer layer, and a clustering layer. Moreover, the idea of adversarial learning is also incoporated into the proposed approach to help the representation learning. Empirical evaluations on three spatiotemporal datasets are provided, showing improvements on the listed baseline methods.

**Strengths:**

+ It demonstrates the application potential of deep subspace clustering techniques in the task of spatiotemporal data segmentation.
+ Experimental results demonstrate the superior performance.

**Weaknesses:**

- It is not clear whether the distribution of the learned embeddings are able to form a union of subspaces.
- The computation complexity and the computational cost of the proposed approach are not clear.
- There are many hyper-parameters, e.g., \alpha, \lambda_bar, \lambda_SE, \lambda_adv. How to set these parameters and what about the sensitivity of the proposed approach with respect these parameters are unkown.
- The empirical evaluation is insufficient. The listed baseline algorithms are out-of-date. It is not clear whether the proposed approach could yield superior performance when comparing to the newly proposed deep subspace or manifold clustering algorithms, e.g., MLC[b], CPP[c], PRO-DSC[d].

Ref:
[a] Unsupervised manifold linearizing and clustering, ICCV'23.

[b] Image clustering via the principle of rate reduction in the age of pretrained models, ICLR'24.

[c] Exploring a Principled Framework for Deep Subspace Clustering, ICLR'25.

**Questions:**

- The reviewer would like to see some evidence of the proposed approach indeed being able to learn stacked embeddings of a union of subspaces.
- It is not clear whether the deep representation learning module suffers from the catestrophic collapse (as illustrated in [a] for DSCNet).
- The summary of the statistics of the used data is not provided.  The computation cost and computation complexity of the proposed approach are not clear. The reviewer is wondering if the proposed approach could scale to large size data.
- How to set these parameters and what about the sensitivity of the proposed approach with respect these parameters are unkown.
- It is not clear whether the proposed approach could yield superior performance when comparing to the newly proposed deep subspace or manifold clustering algorithms, e.g., MLC, CPP, PRO-DSC.

Ref:
[d] A critique of self-expressive deep subspace clustering, ICLR'21.

---

### Official Review · Reviewer_Tpu8 · 2025-10-30

**Soundness:** 2
**Presentation:** 3
**Contribution:** 2
**Rating:** 2
**Confidence:** 4

**Summary:**

The paper introduces a novel model called Attention-guided Deep Adversarial Temporal Subspace Clustering (A-DATSC), which addresses the limitations of traditional subspace clustering models, particularly those based on shallow autoencoders. These models struggle with challenges such as long-range dependencies, local feature extraction, and clustering errors. The A-DATSC model improves upon these shortcomings by combining a deep subspace clustering generator with a quality verification discriminator, utilizing Graph Attention Transformers and ConvLSTM2D layers for enhanced representation learning. The main contributions of the paper are:

1.End-to-End Adversarial Temporal Subspace Clustering Architecture for 4D spatiotemporal data: The paper proposes a new architecture for unsupervised clustering of multi-dimensional, multi-variate spatiotemporal data that effectively handles complex time-series relationships.

2.Graph Attention Transformer for Self-Expression Learning: The model incorporates Graph Attention Transformers to capture both short-range and long-range spatiotemporal dependencies, enhancing the model's ability to represent complex temporal dynamics.

3.Energy-based Temporal Discriminator: The authors introduce a temporal discriminator based on energy, which validates the quality of potential subspaces by reconstructing residuals. This process helps in ensuring the reliability of the learned subspaces.

**Strengths:**

1.The topic is of significant practical relevance, closely addressing the growing application needs of multi-variable spatiotemporal data, and solving key challenges in current science and engineering fields.

2.The model design involves several complex components, but the overall description is clear, and the methodology is relatively complete. The authors conducted extensive testing on multiple datasets and compared the results with mainstream baseline methods, achieving convincing experimental outcomes.

3.The methodology demonstrates a high level of detail and systematization, particularly in the introduction of an energy-based discriminator for adversarial regularization. This is a clever mechanism that explicitly constrains the geometric structure in the latent space, enhancing the separation between different subspaces and effectively improving clustering performance. This innovative design provides a stronger theoretical foundation and practical value for the model.

**Weaknesses:**

1.Limited Innovation: Although the A-DATSC model achieves some performance improvements compared to existing methods in experiments, its core architecture primarily consists of the integration and fine-tuning of multiple existing modules, essentially stacking network structures. It lacks new methodological designs or theoretical breakthroughs, so its innovation is relatively limited.

2.Potentially High Model Complexity: The model includes several sub-networks, such as the generator, encoder, and discriminator, making the overall architecture complex with a large number of parameters and high training costs. While this complexity helps improve performance, it also introduces the risk of overfitting, especially on small-scale spatiotemporal datasets.

3.Limitations of Ablation Studies: The current ablation experiments primarily demonstrate the impact of individual components but do not directly test the effects of combining multiple modules. Given that the model involves several high-complexity components (e.g., ConvLSTM2D, GAT, and Discriminator), these modules may interact differently when used together. As a result, the lack of evaluation of the combined effects of these modules makes the results less complete.

**Questions:**

1.Is it possible to provide more original contributions from a methodological or theoretical perspective?

2.The proposed model in the paper has a complex architecture with several high-complexity modules (such as ConvLSTM2D, discriminator, etc.). Could the authors provide an analysis of key computational efficiency metrics, such as training time, inference speed, GPU memory usage, and the number of parameters? These data would be helpful for evaluating the model's feasibility and performance in practical applications, especially on large-scale datasets.

3.The ablation study in the paper demonstrates the effect of individual modules but lacks an in-depth exploration of the combination effects of the modules. Given that the model involves several high-complexity components, could the authors further test the combined effects of multiple modules?

---

### Official Review · Reviewer_bNFr · 2025-11-01

**Soundness:** 2
**Presentation:** 2
**Contribution:** 2
**Rating:** 2
**Confidence:** 4

**Summary:**

This paper proposes a deep adversarial model for clustering 4D multivariate spatiotemporal data. The method integrates a ConvLSTM-based encoder, a bidirectional temporal graph attention (Bi-TGAT) bottleneck, and a self-expressive temporal layer to capture both local and global dependencies, guided by an energy-based discriminator to enforce subspace geometry. Experiments on three reanalysis climate datasets show that A-DATSC outperforms several existing deep clustering methods in internal clustering metrics.

**Strengths:**

- Subspace clustering for multidimensional multivariate spatiotemporal data has wide applicaitons and addressing the challenges in this problem is necessary.

- The proposed method ahieves strong performance across multiple real-world climate datasets, outperforming prior deep subspace clustering methods on several metrics.

**Weaknesses:**

- The technical novelty is limited. The model primarily combines existing elements (ConvLSTM, attention, adversarial subspace learning, and DEC) rather than introducing fundamentally new mechanisms.

- There is insufficient discussion of how this methods differ or improve over existing adversarial subspace clustering methods, e.g. DASC and DSC-DAG. There are a lot of descriptions of model arthictecture, e.g. network design, cluster head, etc. However the insight of why the specific design is superior is missing.

- Most comparisons are against methods before 2020, omitting more recent transformer-based or contrastive deep clustering models.

- Evaluation is only conducted on climate datasets. Thus it might be better to highlight the focus in the title by explicitly stating climate prediction.

- The experiments presetation is poor. For example, the caption of Tab 1 indicates best result is underlined which is, however, not consistent with the contents in the table. Moreover, some rows in Tab 1 has no boldface number.

**Questions:**

- Provide more insight into the methodology

- Include additional comparisons with more recent methods.

---

### Note · Authors · 2025-12-09

I have read and agree with the venue's withdrawal policy on behalf of myself and my co-authors.